# Mathematical Analysis and Motion Capture System Utilization Method for Standardization Evaluation of Tracking Objectivity of 6-DOF Arm Structure for Rehabilitation Training Exercise Therapy Robot

**DOI:** 10.3390/diagnostics12123179

**Published:** 2022-12-15

**Authors:** Jaehwang Seol, Kicheol Yoon, Kwang Gi Kim

**Affiliations:** 1Department of Biomedical Engineering, College of Health Science, Gachon University, 191 Hambak-Moero, Yeonsu-gu, Incheon 21936, Republic of Korea; 2Medical Devices R&D Center, Gachon University Gil Medical Center, 21, 774 Beon-gil, Namdong-daero, Namdong-gu, Incheon 21565, Republic of Korea; 3Department of Biomedical Engineering, College of Medicine, Gachon University, 38-13, 3 Beon-gil, Dokjom-ro 3, Namdong-gu, Incheon 21565, Republic of Korea; 4Department of Health Sciences and Technology, Gachon Advanced Institute for Health Sciences and Technology (GAIHST), Gachon University, 38-13, 3 Beon-gil, Dokjom-ro, Namdong-gu, Incheon 21565, Republic of Korea

**Keywords:** shoulder pain, rehabilitation robot, motion capture system, inverse kinematics, range of motion, standardization evaluation

## Abstract

A treatment method for suppressing shoulder pain by reducing the secretion of neurotransmitters in the brain is being studied in compliance with domestic and international standards. A robot is being developed to assist physical therapists in shoulder rehabilitation exercise treatment. The robot used for rehabilitation therapy enables the training of patients to perform rehabilitation exercises repeatedly. However, the biomechanical movement (or motion) of the shoulder joint should be accurately designed to enhance efficiency using a shoulder rehabilitation robot. Furthermore, safely treating patients by accurately evaluating biomechanical movements in compliance with domestic and international standards is a major task. Therefore, an in-depth analysis of shoulder movement is essential for understanding the mechanism of shoulder rehabilitation using robots. This paper proposes a method for analyzing shoulder movements. The rotation angle and range of motion (ROM) of the shoulder joint are measured by attaching a marker to the body and analyzing the inverse kinematics. The first motion is abduction and adduction, and the second is external and internal rotation. The location information of the marker is transmitted to an application software through an infrared camera. For the analysis using an inverse kinematics solution, five males and five females participated in the motion capture experiment. The subjects did not have any disability, and abduction and adduction were repeated 10 times. As a result, ROM of the abduction and adduction were 148° with males and 138.7° in females. Moreover, ROM of the external and internal rotation were 111.2° with males and 106° in females. Because this study enables tracking of the center coordinates of the joint suitably through a motion capture system, inverse kinematics can be accurately calculated. Additionally, a mathematical inverse kinematics equation will utilize follow-up study for designing an upper rehabilitations robot. The proposed method is assessed to be able to contribute to the definition of domestic and international standardization of rehabilitation robots and motion capture for objective evaluation.

## 1. Introduction

Motor nerves transmit signals from the brain to muscles to induce movement of the shoulder and arm. In particular, when shoulder pain is induced, the muscle can be relaxed by physically stimulating it to relieve pain. Therefore, research is being conducted in compliance with domestic and international standards (IEC 80601-2-78:2019 and SC43) to suppress shoulder pain by reducing the neurotransmitter secretion in the brain. Shoulder pain is a common complication that can be caused by adhesive capsulitis and hemiplegia induced by a stroke [1]. In particular, the adhesive capsulitis causes shoulder pain due to the thickening of the joint capsule and the adhesion of tendons or ligaments [1]. Adhesive capsulitis also causes additional complications due to rotator cuff tears. Therefore, shoulder pain can be reduced through stretching and passive and active joint exercise treatment [1].

Shoulder pain in hemiplegia and adhesive capsulitis requires nonsurgical treatment and shoulder rehabilitation (SR). Rehabilitation exercises have been enabled through conventional manual therapy by physical therapists. However, owing to the development of biomedical engineering technology, the research and development of medical robots for rehabilitation treatment continues through the convergence of physical therapy and engineering [2,3,4,5,6,7]. The advantage of a rehabilitation robot is that therapists are able to train the patient, such that a male or female can repeatedly perform rehabilitation exercises [8,9]. The safety requirements of robots for rehabilitation exercise therapy are extremely important, as specified in the international standards (IEC 80601-2-78:2019). A representative requirement of international standardization of the safety of robots for rehabilitation exercise therapy is that when a hemiplegic or speech-impaired person is trained in a robot system to receive SR, communication between the therapist and the patient must be established [8]. However, it is difficult for a patient with a disability to convey meaning to the therapist, and if an emergency occurs, the paralyzed person must deliver a message to the therapist. However, it is difficult for these patients to convey a clear message. Therefore, these problems lead to medical accidents, making it necessary to establish domestic and international standardization of computer interfaces through which patients and therapists can communicate. Consequently, it is necessary to introduce an intelligent rehabilitation treatment robot to be able to deliver a message in an emergency and monitor the patient’s condition.

In addition, the characteristics of the SR robot enable repetitive exercise training through the automation system, reducing the fatigue of the therapist who needs to perform extensive work, and can guide SR exercise training more accurately [9,10]. However, it is important to accurately implement the biomechanical movement (or motion) of the shoulder joint to enhance the efficiency of using a shoulder rehabilitation robot. The accurate movement of the SR robot can ensure patient safety and prevent accidents [9,10]. Therefore, an in-depth analysis of shoulder movement is essential for understanding the mechanism of SR robots. Various studies on the mechanism of shoulder movement have been conducted [11,12,13,14,15,16].

Wu et al., from the International Society of Biomechanics, proposed a shoulder model based on the definition of the shoulder joint coordinate system (JCS). In particular, the proposed method presented the standardization of the JCS for the shoulder, elbow, wrist, and hand [14], thereby contributing to smooth communication between researchers and clinicians regarding kinematics. However, during the repetitive experiment, the standard position of the joint is not constant and has limitations [14]. Jackson et al. analyzed shoulder kinematics by attaching a marker to the skin to fix the standard joint position. In particular, the method using the chain model and Kalman filter reconstructs the shoulder kinematics by tracking the trajectory of the marker. Therefore, the burden is reduced to an extent that it is unnecessary for the reconstruction of the mathematical model for the determination of the range of motion (ROM) [15]. Zhang et al. proposed a kinematic model using a Vicon motion capture system and markers. In particular, the shoulder elevation and depression phases, and the movement coupling relationship between the displacement of the glenohumeral (GH) joint center with respect to the thoracic coordinate system and elevation of the humerus was investigated. As a result, a new design model for an upper extremity rehabilitation robot consistent with the actual situation of the human body structure was developed [16].

Similar to previous studies, this study proposes a method for analyzing shoulder movements. The rotation angle and ROM of the shoulder joint were measured by attaching a marker to the body and analyzing the inverse kinematics. In particular, a rigid body was designated through a marker to accurately determine the internal center point of the joint. For the experiment, subjects of this study (five males and five females) without any functional disability in the body participated in the motion capture test. Based on the information, which was obtained by tracking the position of the marker, the ROM of each joint was analyzed using inverse kinematics. Consequently, motion analysis using inverse kinematics will be applied to the mechanism of rehabilitation robots. In addition, ROM information of a normal subject can be used as a database for utilizing an SR robot for rehabilitation exercises.

## 2. Analysis of Motion Capture

In the process of using the robot system for rehabilitation-based training treatment, patients receiving treatment for shoulder pain disease with hemiplegia or speech impairment can communicate with the therapist using a computer, as shown in Figure 1a [8].

Quadriplegic, deaf, blind, and speech-impaired patients cannot express themselves accurately to therapists during exercise training programs for rehabilitation treatment [17]. Therefore, if emergencies occur during the course of training and treatment using treatment instruments, the therapist may not recognize the patient’s condition and a medical accident can occur. Brain computer interface (BCI) defines a technology for interaction between the brain and a computer [18]. This technology refers to a control technology that provides a service so that a computer can grasp the thoughts intended by humans and move objects [19]. In other words, BCI detects brain waves so that computers can grasp cognition, learning, and reasoning similarly to the human brain [20]. Therefore, it is predicted that the use of BCI technology will be high for quadriplegic, hearing-impaired, visually-impaired, and speech-impaired patients who need rehabilitation exercise. BCI technology uses a camera to capture the movement of the patient, and accurately reads an EEG from the patient. It then analyzes the data obtained from the camera and EEG diagnosis to identify the patient’s movement pattern. Therefore, it is possible to predict the treatment outcome by understanding the patient’s requirements and condition.

It is desirable to use a robotic system in which such brain-computer interface (BCI) standardization (SC43) has been established. The most important aspect when moving the arm of the robot in the process of robot motion is matching the movement of the patient’s shoulder. Therefore, an objective evaluation is important to match the patient’s shoulder movement when the robot’s arm moves, and domestic or international standardization work for this evaluation method is highly important [8]. In considering the movement of the robot arm and patient shoulder to establish standardization, it is important to study the construction of a motion capture-based monitoring system for objective evaluation and a mathematical algorithm analysis method for verifying the objective evaluation. In this way, it is possible to provide a safe rehabilitation robot therapy (IEC 80601-2-78:2019) to patients. 

Figure 1b shows the setup environment for the motion capture experiment. The overall movement, such as position data of the arm, was tracked through motion capture, and the value of the end effector was obtained. In this study, the wrist was designated as an end effector and utilized as input data to interpret the inverse kinematics. Accordingly, the position and direction vectors of the wrist were tracked in real time through the motion capture system.

The subjects wore stretchy suits to demonstrate that the markers could be attached to the skin. The markers were coated with a material that reflects infrared light, which transmits the position data of the markers to the application software (Motive) using an infrared camera (Flex13, OptiTrack). Consequently, the position vector and direction vector of the markers were extracted in real time based on the absolute coordinate system in the software. In this study, the position data of the markers were analyzed by tracking the two rehabilitation motions. The first motion is abduction and adduction, and the second is external and internal rotation.

Figure 2 shows the location of the markers that were attached to the elastic suit. As shown in Figure 2a, the markers were attached to the clavicle, shoulder, elbow, and wrist. The joints of the arms are located internally and contribute to the rotation of the bones. Therefore, the markers were attached with the center position coinciding with the internal center of the joint. While attaching the markers to designate the subjects’ joint center points, the accuracy was increased by attaching the markers with help of an on-site physical therapist.

Figure 2b shows the locations of the marker attachments and central coordinates of the bone structure of the right arm. The sternoclavicular (SC) protrudes because the muscular membrane and skin covering the joint are thinner than other areas of the body. Therefore, one marker was attached without calculating the central coordinate. Three markers were attached to the shoulder to designate the glenohumeral (GH) joint as the central coordinate system. Two markers were attached to the elbow and wrist, and the humeroulnar (HU) joint and distal radioulnar (DRU) joint were designated as the center coordinates. 

## 3. Mechanism and Mathematical Analysis

### 3.1. Forward Kinematics

Before interpreting an inverse kinematics solution, forward kinematics was analyzed and defined as a homogeneous transformation matrix [21]. Figure 3 shows the forward kinematics modeling of the right arm that is expressed based on the rotation joint.

Figure 3a shows the rotation joints contributing to the movement of the arm at each central joint position. In particular, points O, S, E, and EE (indicated by the blue dashed circle) are the center points of the joint coordinate system and represent the center coordinates of the joint rotation designated through motion capture. Point O (SC joint) comprises a two-axis rotation joint that involved the vertical and horizontal rotation of the clavicle. Point O is designated as the base point in the kinematics model. Point S (GH joint) is composed of three-axis rotation joints that involved the roll, pitch, and yaw rotation. Point E (HU and HR joint) is composed of a uniaxial rotation joint that involved the flexion and extension of the arm. Finally, point EE (DRU joint) is designated as the end effector of the forward kinematics. In the following kinematics analysis process, the central coordinates of the clavicle, shoulder, elbow, and wrist are expressed as points O, S, E, and EE, respectively.

Figure 3b shows the forward kinematics model of the shoulder with the moving coordinate system. In each joint, the $X_{i}$, $Y_{i}$, and $Z_{i}\;{(}_{i=0\;\mathrm{to}\;6})$ axes that are the movement coordinate systems were mapped to the joint $\theta_{i}$. The links and rotation parameters based on the forward kinematics are shown in Table 1 and were determined from the Denavit–Hartenberg proof [22,23]. In particular, $\theta_{i}$ is the rotation joint and directly concerns the rehabilitation exercise. Therefore, it is an important to measure $\theta_{i}$ and ROM in this study. 

The link offset and length (e.g., humerus or radius) are from different subjects. Therefore, the links can be calculated through the distance formula between two points in 3-dimensional space to substitute inverse kinematics as a constant value. Equation (1) represents the distance formula of links ($d_{i}$ or $l_{i}$) based on the arbitrary 3-dimensional position from the $X_{n},\;Y_{n}$ and $Z_{n\;\left(n=natural\;number\right)}$ position. To reflect the links that change in real time in the forward and inverse kinematics, a MATLAB tool was used.
(1)$$l_{i}=d_{i}=\sqrt{{\left(X_{i}-X_{i-1}\right)}^{2}+{\left(Y_{i}-Y_{i-1}\right)}^{2}+{\left(Z_{i}-Z_{i-1}\right)}^{2}}$$
(2)T 01=C10−S10S10C100−1000001, T 12=C20S2l2C2S20−C2l2S201000001, T 23=C30−S30S30C300−1000001 ,
T 34=C40S40S40−C4001000001, T 45=C50−S50S50C500−10d50001, T 56=C6−S60l6C6S6C60l6S600100001

Based on the information in Table 1, a homogeneous transformation matrix of each rotation joint is shown in Equation (2). Among the components of the matrix, the 3 × 3 matrix (row: 1 to 3, column: 1 to 3) represents the rotation matrix, and the 3 × 1 matrix (row: 1 to 3, column: 4) represents the position vector.
(3)T 06=T 01T 12T 23T 34T 45T 56=R11R12R13PxR21R22R23PyR31R32R33Pz0001
(4)T 05=T 01T 12T 23T 34T 45=r11r12r13Xer21r22r23Yer31r32r33Ze0001

Equation (3) represents the multiplication of the matrix from points O to EE. The direction vectors are expressed as $R_{\mathrm{ij}\;\left(i,j=1\;\mathrm{to}\;3\right)}$ and the position vectors are expressed as $P_{i\;\left(i=x,\;y,\;z\right)}$. Equation (4) represents the multiplication of the matrix from point O to point E. Similarly, the direction vectors are included as $r_{\mathrm{ij}\;\left(i,j=1\;\mathrm{to}\;3\right)}$ and the position vectors are included as $I_{e\;\left(I=X,\;Y,\;Z\right)}$.

### 3.2. Inverse Kinematics

#### 3.2.1. Position Vector Analysis

The end effector is defined as a homogeneous transformation matrix through motion capture. Subsequently, the position vector of the elbow is calculated utilizing the end effector data. Figure 4 shows the position and direction vector of each point. As shown in Equation (5), the position vector of point E ($X_{e}$, $Y_{e}$, $Z_{e}$) is calculated through the x-axis direction vector of the end effector and link $l_{6}$.
(5)$$\mathrm{EE}=\left[\begin{array}{c} P_{x}\\ P_{y}\\ P_{z} \end{array}\right],\;E=\mathrm{EE}-l_{6}R\left[\begin{array}{c} 1\\ 0\\ 0 \end{array}\right]=\left[\begin{array}{c} P_{x}-l_{6}R_{11}\\ P_{y}-l_{6}R_{21}\\ P_{z}-l_{6}R_{31} \end{array}\right]$$

In particular, $R_{\mathrm{ij}\;\left(i,j=1\;\mathrm{to}\;3\right)}$ represents the rotation matrix of the end effector. Therefore, the direction vector of the x-axis is analyzed by multiplying the transposition matrix ${\left[1\;0\;0\right]}^{T}$ with the R matrix, and the links ($l_{6}$) are multiplied to calculate the magnitude of the x-axis direction. Consequently, the position vector of point E ($X_{e},\;Y_{e},\;Z_{e}$) is calculated by subtracting, as shown in Equation (5).
(6)$$\vec{\mathrm{ES}}=\vec{\mathrm{OS}}-\vec{\mathrm{OE}}=\langle X_{c}-X_{e},Y_{c}-Y_{e},Z_{c}-Z_{e}\rangle$$
(7)$$\vec{R_{z}}=\langle R_{13},R_{23},R_{33}\rangle$$
(8)$$\vec{\mathrm{ES}}\cdot \vec{R_{z}}=\left|\vec{\mathrm{ES}}\cdot \vec{R_{z}}\right|\cdot \cos\frac{\pi }{2}=0$$
(9)$$\vec{\mathrm{ES}}\cdot \vec{{\mathrm{EO}}_{\;}}=\left|\vec{\mathrm{ES}}\cdot \vec{{\mathrm{EO}}_{\;}}\right|\cdot {\cos\theta }_{0}=\langle X_{c}-X_{e},Y_{c}-Y_{e},Z_{c}-Z_{e}\rangle \cdot \langle -X_{e},-Y_{e},-Z_{e}\rangle$$
(10)$$R_{13}X_{c}+R_{23}Y_{c}+R_{33}Z_{c}=\alpha \;,\left(\alpha =R_{13}X_{e}+R_{23}Y_{e}+R_{33}Z_{e}\right)$$
(11)$$X_{e}X_{c}+Y_{e}Y_{c}+Z_{e}Z_{c}=\beta \;,(\beta =X_{e}{}^{2}+Y_{e}{}^{2}+Z_{e}{}^{2}-\vec{\left|\mathrm{ES}\right|}\cdot \vec{\left|\mathrm{EO}\right|}\cdot {\cos\theta }_{0}$$
(12)$${\cos\theta }_{0}=\frac{d_{5}{}^{2}+(X_{e}{}^{2}+Y_{e}{}^{2}+Z_{e}{}^{2})-l_{2}{}^{2}}{2\cdot d_{5}\cdot \sqrt{X_{e}{}^{2}+Y_{e}{}^{2}+Z_{e}{}^{2}}}$$

In Equation (6), the $\vec{\mathrm{ES}}$ vector is calculated by subtracting the vectors $\vec{\mathrm{OS}}$ and $\vec{\mathrm{OE}}$. In Equation (7), the vector $\vec{R_{z}}$ is defined as the z-axis direction vector of the end effector. Equations (8) and (9) show the dot product formula between vectors $\vec{\mathrm{ES}}$ and $\vec{\mathrm{EO}}$. As shown in Equation (8), vectors $\vec{\mathrm{ES}}$ and $\vec{R_{z}}$ are always perpendicular, and the magnitude of the dot product always converges to zero. Equation (9) shows the left and right mathematical expression that represent the identities. Equations (8) and (9) can be induced and arranged into Equations (10) and (11). In particular, α and β are substituted variable values for constant value through the position and direction vector of EE and E. Consequently, ${\cos\theta }_{0}$ is obtained by calculating the internal angle through $\vec{\mathrm{ES}}$ and $\vec{\mathrm{EO}}$ in ΔOSE.
(13)$$(R_{23}-\frac{Y_{e}}{X_{e}}R_{13})Y_{C}+(R_{33}-\frac{Z_{e}}{X_{e}}R_{13})Z_{C}=\alpha -\frac{R_{13}}{X_{e}}\beta \;\to \;p_{1}Y_{c}+q_{1}Z_{c}=r_{1}$$
(14)$$(R_{13}-\frac{X_{e}}{Y_{e}}R_{23})X_{C}+(R_{33}-\frac{Z_{e}}{Y_{e}}R_{23})Z_{C}=\alpha -\frac{R_{23}}{Y_{e}}\beta \;\to \;p_{2}X_{c}+q_{2}Z_{c}=r_{2}$$

Equations (10) and (11) are combined and expressed as a simultaneous equation and induced to Equations (13) and (14). In particular, the argument of $X_{C}$, $Y_{C}$, $Z_{C}$, and right mathematical expression are defined as constant values in Equations (5)–(12). Therefore, $p_{1}$, $q_{1}$, and $r_{1}$ are respectively defined as variable values of $X_{C}$, $Y_{C}$, and $Z_{C}$ in Equation (13). Similarly, Equation (14) defines the variable value as $p_{2}$, $q_{2}$, and $r_{2}$.
(15)$$X_{c}{}^{2}+Y_{c}{}^{2}+Z_{c}{}^{2}=l_{2}{}^{2}$$
(16)$$(\frac{q_{1}{}^{2}}{p_{1}{}^{2}}+\frac{q_{2}{}^{2}}{p_{2}{}^{2}}+1)Z_{C}{}^{2}-2(\frac{q_{1}r_{1}}{p_{1}{}^{2}}+\frac{q_{2}r_{2}}{p_{2}{}^{2}})Z_{C}+(\frac{r_{1}{}^{2}}{p_{1}{}^{2}}+\frac{r_{2}{}^{2}}{p_{2}{}^{2}})=l_{2}{}^{2},\;(Z_{C}>0)$$

Equation (15) is the equation of a sphere that has center point from point O. The distance between points S and O represents the radius of the sphere and is equal to link $l_{2}$. Therefore, by substituting Equations (13)–(15), Equation (16) can be expressed as a quadratic equation for $Z_{C}$.

Figure 5 shows the mathematical relationship between Equations (10), (11), (15), and (16) in 3D coordinate space. It is possible to geometrically interpret a quadratic equation that $Z_{C}$ is a variable. In particular, Equations (10) and (11) are presented by a three-dimensional plane. Therefore, the two planes are crossed and make an intersection line, and the intersection line passes through the sphere to obtain the two intersection points. Consequently, the two intersection points have a potential to be solutions of Equation (16), being the z-axis position vector of point S.

Two solutions are obtained in Equation (16). According to the joint structure of the upper limb, one solution is selected by considering the normal biomechanical movement. Figure 6 shows the biomechanical relationship between the shoulder and the acromioclavicular joint. In Figure 6a, the head of the humerus is covered by the glenohumeral joint and the subacromial bursa. The head of the humerus relaxes or contracts through the supraspinatus and becomes the axis of shoulder rotation. Simultaneously, with the rotation of the shoulder, the clavicle rotates through the sternal end that becomes the axis of rotation. Therefore, the rotary direction of the shoulder and clavicle are the same, as shown in the normal state in Figure 6b. In contrast, the rotation of the shoulder and clavicle are in opposite directions in the abnormal state shown in Figure 6b. Therefore, the movement of the shoulder has the potential to create friction between the humeral head and the acromion. 

Two solutions of Equation (16) determine $Z_{C}$ as the position vectors of point S. According to biomechanical analysis, the calculation of Equation (16) can add two conditions. A comparison is possible when it is assumed that two $Z_{C}$ values are expressed as $Z_{C1}$ and $Z_{C2}$. If $Z_{C2}$ > $Z_{C1}$ and $Z_{C1}$ is selected as the solution, the center coordinate of the shoulder is always located below the horizontal line. Therefore, the clavicle has a downward oblique angle and an abnormal state, as shown in Figure 6b. In contrast, if $Z_{C2}$ is selected as the solution, point S is located above the horizontal line. Therefore, the clavicle maintains the upper oblique angle and a normal state, as shown in Figure 6a. As a result, a condition is ensured to select $Z_{C2}$ when the condition is added, such as $Z_{C2}>0>Z_{C}$.

Based on Equations (13) and (14), the position values of $X_{C}$ and $Y_{C}$ were calculated using the selected $Z_{C}$. The head of the humerus is attached to the acromion and fixed by the pectoralis major, supraspinatus, and infraspinatus. Therefore, when determining $X_{C}$, the condition $X_{C}$ > 0 is ensured, based on point O (sternoclavicular). As a result, when determining $Z_{C}$, the conditions that $Z_{C2}>0>Z_{C}$ and $X_{C}$ > 0 can be added.

Figure 7 shows the results when the conditions ($Z_{C}$ > 0 and $X_{C}$ > 0) are violated by the simulation (Robo analyzer). The position and direction vector of the end effector are inputted, and the angle of the rotation joint is calculated. In abduction, the $Z_{C}$ and $X_{C}$ values are negative, causing shoulder dislocation, as shown in Figure 7A. Similarly, if $Z_{C}$ is negative during external rotation, shoulder dislocation occurs, as shown in Figure 7B. In summary, the position vector of points EE, E, and S are calculated by adding appropriate conditions. Based on the proper position vector, the angle of the rotation joint will be obtained. 

#### 3.2.2. Joint Angle Analysis

The joint rotation angles are analyzed to calculate the ROM of each rehabilitation motion. In particular, the inverse kinematics solution of the 6-degree of freedom (DOF) is obtained by solving the position vectors of points E and S in advance [19]. This study used the Mathematica tool (Wolfram Alpha) to solve complex trigonometric functions. In this section, ${\cos\theta }_{n}$ and $\sin\theta_{n}$ are replaced by the $C_{n}$ and $S_{n}$ (n = positive number).

Equations (17) and (18) show the calculation process for the joint angle $\theta_{1}$. In Equation (17), $X_{C}$ and $Y_{C}$ are the position vectors of point S. In particular, because the coordinate of one point is included in the spherical coordinate system, $X_{C}$ and $Y_{C}$ are expressed as $l_{2}$, $C_{1}$, $C_{2}$, and $S_{1}$. Therefore, $\theta_{1}$ is calculated by dividing the two position vectors. Arctan2 is used to consider the sign of the angle.
(17)$$X_{C}=l_{2}C_{2}C_{1},{\;Y}_{C}=l_{2}C_{2}S_{1}$$
(18)$$\theta_{1}=\mathrm{atan}2\left(Y_{C},{\;X}_{C}\right)$$

Equations (19)–(21) show the calculation process for the joint angle $\theta_{2}$. Because the left and right mathematical expressions of Equation (19) constitute the same homogeneous transformation matrix, both sides of the matrix have equal element values. Therefore, Equations (20) and (21) are derived through the comparison of the element (row: 1 column: 4) and (row: 2 column: 4) by the homogeneous transformation matrix. As a result, $\theta_{2}$ is calculated through dividing $l_{2}S_{2}$ and $l_{2}C_{2}$. Similar to the calculation process for $\theta_{2}$, the remaining joint angle is solved by comparing the element from both sides of the homogeneous transformation matrix.
(19)T 01−1·T 02=T 12
(20)$$C_{1}X_{C}+S_{1}Y_{C}=l_{2}C_{2}$$
(21)$$-Z_{C}=l_{2}S_{2}$$
(22)$$\theta_{2}=\mathrm{atan}2\left(-Z_{C},{\;C}_{1}X_{C}+S_{1}Y_{C}\right)$$

In Equation (23), both sides of the element values of (row: 1 column: 4) and (row: 2 column: 4) are compared. Equations (24) and (25) are the left element equation and are substituted with characteristics such as a and b. Subsequently, characteristics a and b are multiplied by $C_{3}$ and $S_{3}$ to derive Equation (26), which is expressed in a simultaneous equation with Equations (27) and (28). Similarly, both sides of the element values of (row: 1 column: 1) and (row: 2 column: 1) are compared. Equations (27) and (28) are the left element equation and are substituted with c and d. After respectively multiplying c and d by $C_{3}$ and $S_{3}$, Equation (29) can be expressed through a simultaneous equation. As a result, Equations (26) and (29) are pressed by comparing both sides of the element and divided to derive $\theta_{3}$.
(23)T 02−1·T 06=T 26
(24)$$a=C_{1}C_{2}P_{x}+C_{2}S_{1}P_{y}-S_{2}P_{z}-l_{2}$$
(25)$$b=-S_{1}P_{x}+C_{1}P_{y}$$
(26)$$l_{6}C_{6}S_{5}=-aS_{3}+{\mathrm{bC}}_{3}$$
(27)$$c=C_{1}C_{2}R_{11}+C_{2}S_{1}R_{21}-S_{2}R_{31}$$
(28)$$d=-S_{1}R_{11}+C_{1}R_{21}$$
(29)$$C_{6}S_{5}=-cS_{3}+{\mathrm{dC}}_{3}$$
(30)$$\theta_{3}=\mathrm{atan}2\left(b-l_{6}d,\;a-l_{6}c\right)$$

In Equation (31), the element values of (row: 1, column: 3) and (row: 2 and column: 3) are compared. The left and right mathematical expression of the matrix element are replaced by P and Q, as shown in Equations (32) and (33). As a result, $\theta_{4}$ is calculated by dividing Q and P.
(31)T 03−1·T 06=T 36
(32)$$P=\left(C_{1}C_{2}C_{3}-S_{1}C_{3}\right)R_{13}+\left(C_{2}C_{3}S_{1}+C_{1}S_{3}\right)R_{23}-C_{3}S_{2}R_{33}=-C_{4}S_{5}$$
(33)$$Q=-C_{2}S_{2}R_{13}-S_{1}S_{2}R_{23}-C_{2}R_{33}=-S_{4}S_{5}$$
(34)$$\theta_{4}=\mathrm{atan}2\left(Q,\;P\right)$$

In the left mathematical expression of Equation (35), the element values of (row: 1, column: 1), (row 1, column 2), and (row 1, column 3) are substituted with α, β, and γ, respectively. On the right side, (row 2, column 1), (row 2, column 2), and (row 2, column 3) are substituted with a, b, and c, respectively. As a result, Equations (36) and (37) are divided to calculate $\theta_{5}$.
(35)T 04−1·T 06=T 46
(36)$${\alpha R}_{11}+{\beta R}_{21}+{\gamma R}_{31}=S_{5}C_{6}$$
(37)$${\mathrm{aR}}_{11}+{\mathrm{bR}}_{21}+{\mathrm{cR}}_{31}=C_{5}C_{6}$$
(38)$$\theta_{5}=\mathrm{atan}2\left({\alpha R}_{11}+{\beta R}_{21}+{\gamma R}_{31},{\;\mathrm{aR}}_{11}+{\mathrm{bR}}_{21}+{\mathrm{cR}}_{31}\right)$$

Finally, $\theta_{6}$ compares the element values of (row 1, column 1) and (row 2, column 1) in the left and right terms of Equation (39). In matrix T 05−1, (row 1, column 1), (row 1, column 2), and (row 1, column 3) are replaced by $U_{1}$, $U_{2}$, $U_{3}$, respectively. Additionally, (row 2, column 1), (row 2, column 2), and (row 2, column 3) are replaced by $V_{1}$, $V_{2}$, and $V_{3}$, respectively. Consequently, Equations (40) and (41) are divided to calculate $\theta_{6}$.
(39)T 05−1·T 06=T 56
(40)$$U_{1}R_{11}+U_{2}R_{21}+U_{3}R_{31}=C_{6}$$
(41)$$V_{1}R_{11}+V_{2}R_{21}+V_{3}R_{31}=S_{6}$$
(42)$$\theta_{6}=\mathrm{atan}2\left(V_{1}R_{11}+V_{2}R_{21}+V_{3}R_{31},{\;U}_{1}R_{11}+U_{2}R_{21}+U_{3}R_{31}\right)$$

## 4. Experiment Results and Discussion

### 4.1. Abduction and Adduction

Prior to the analysis, five randomized males and five randomized females participated in the motion capture experiment. The subjects did not show any disability. Abduction and adduction motions were repeated 10 times. Figure 8 shows the joint rotation angle, ROM, and simulation results from abduction and adduction.

Figure 8a shows the joint rotation pattern of a subject who performed the abduction and adduction. While each subject performed the exercise 10 times, the similar pattern of the joint angle appeared from $\theta_{1}$ to $\theta_{6}$. In particular, the shoulder joint ($\theta_{4}$) has the largest variation degree. Simultaneously, the clavicle joint ($\theta_{2}$) rotates in the same direction with $\theta_{4}$. All subjects have different ROM, and the quantitative ROM information is listed in Table 2.

Table 2 shows the ROM of males (M) and females (F) in abduction and adduction. The average ROM for the horizontal angle of the clavicle ($\theta_{1}$) was 28.9° and 18.3° for males and females, respectively, and the ROM for the vertical angle of the clavicle ($\theta_{2}$) was 17.6° and 11.5°, respectively. Therefore, both $\theta_{1}$ and $\theta_{2}$ average ROM for males was higher than that of females. Roll ($\theta_{3}$), pitch ($\theta_{4}$), and yaw ($\theta_{5}$) of the shoulder joint contribute to the shoulder rotation. The average ROM of roll ($\theta_{3}$) was 46.1° and 31.9° for males and females, respectively, and yaw ($\theta_{5}$) was 69.3° and 44.8° for males and females, respectively, indicating that the ROM of males was higher than that of females. In particular, the ROM of pitch ($\theta_{4}$) was 130.4° and 127.2°. Therefore, $\theta_{3}$, $\theta_{4}$, and $\theta_{5}$ values for the males were higher than the females. Elbow joint ($\theta_{6}$) was 20.7° and 26.2° for males and females, respectively. As a result, males have higher average ROM in the clavicle and shoulder than females, whereas females have higher average ROM in the elbow. For the average ROM by rotation angle of 10 subjects, the standard deviation (σ) was calculated. The standard deviation of $\theta_{2}$ and $\theta_{4}$ that significantly contributes the abduction and adduction is 4.8 and 10.0. 

Figure 8b graphically shows the average ROM for 10 subjects through an analysis of Table 2. Rounding was performed to the first decimal place. The angles of $\theta_{1}$ and $\theta_{2}$ that contributed to the movement of the clavicle were 24° and 15°, respectively. Moreover, angles of $\theta_{3}$, $\theta_{4}$, and $\theta_{5}$ that are involved in shoulder movement were 39°, 129°, and 57°, respectively. The elbow movement ($\theta_{6}$) has 23° in abduction and adduction. Figure 8c shows the simulation of the abduction and adduction based on the average ROM. The standing motion was set to the initial position that reflects initial angle value in the parameter of Figure 8c. Consequently, robot simulation shows the accurate trajectory that starts from the initial point and end point of the rotation angle with six-axis joints. 

Figure 8d shows the joint-centered trajectory graph in a 6-axis arm structure based on the authors’ motion capture experiment. In the figure, the clavicle maintains a relatively constant position. On the other hand, the shoulder, elbow, and wrist have repetitive movements. Based on the figure, we can analyze the one difference between the simulation and movement of humans in Figure 8c,d. The robot simulation gives a certain angle to form repetitive ROM in one caption line. However, in the human movement based on the motion capture data, the ROM was obtained through repetitive motion in various caption lines, as shown in Figure 8d. The caption line of abduction and adduction changes within 45 degrees and moves to maintain a constant ROM.

### 4.2. External and Internal Rotation

Figure 9 shows the joint rotation angle, ROM, and simulation results from external rotation and internal rotation. 

Figure 9a shows the joint rotation pattern of a subject who performed the external and internal rotation. As with abduction and adduction, each subject performed the exercise 10 times, and the similar pattern of the joint angle appeared from $\theta_{1}$ to $\theta_{6}$. In particular, the shoulder joint ($\theta_{5}$) has the largest degree of variation. Furthermore, all the joints, except the elbow joint (θ_6_), showed relatively small movement. As for abduction and adduction, Table 3 summarizes the ROM for the ten subjects in an external and internal rotation experiment.

The average ROM for the horizontal angle of the clavicle ($\theta_{1}$) was 4.9° and 2.9° for males and females, respectively, and the ROM for the vertical rotation ($\theta_{2}$) was 3.2° and 3.4°, respectively. Therefore, the $\theta_{1}$ degree for males is significantly higher than females. However, degree $\theta_{2}$ for females is significantly higher than males. The average ROM of roll ($\theta_{3}$) was 8.0° and 7.9° for males and females, respectively, and pitch ($\theta_{4}$) was 8.5° and 7.5° for males and females, respectively, indicating that both ROM for males was slightly higher than females. In particular, the average ROM of yaw ($\theta_{5}$) was 111.1° and 106.0°. Therefore, degree $\theta_{5}$ for the males is significantly higher than females. The ROM of the elbow ($\theta_{6}$) was almost same in males and females at 23.4° and 23.6°, respectively. The standard deviation of the average ROM (σ) was calculated, and the standard deviation of $\theta_{5}$, which significantly contributes the external and internal rotation, is 18.2.

Figure 9b graphically shows the average ROM for 10 subjects through an analysis of Table 3. Rounding was performed to the first decimal place. Angle of $\theta_{1}$ and $\theta_{2}$ that contributed to movement of clavicle were 4° and 3°, respectively. Moreover, angle of $\theta_{3}$, $\theta_{4}$ and $\theta_{5}$ that involved in shoulder movement were 8°, 8° and 109°. The elbow movement ($\theta_{6})$ has 24° in external and internal rotation. Figure 9c shows the simulation of the external and internal based on the average ROM. The standing motion was set to the initial position that reflects initial angle value in the parameter of the Figure 8c. Consequently, robot simulation shows the accurate trajectory that starts from initial point and end point of the rotation angle with six-axis joints. Figure 8d shows the joint-centered trajectory graph in 6-axis arm structure. In the figure, clavicle and shoulder maintain a relatively constant position. On the other hand, wrist have repetitive movements with axis of radius. As a result, we analyzed that both the simulation and the subject maintain a constant scription line, repeating the movement. 

### 4.3. Discussion

This study measured the joint angle degree and ROM of the 6-DOF for two SR motions of the subjects. The joint angles ($\theta_{4}$) that are most significantly involved in abduction and adduction were 130.4° and 127.2° for males and females, respectively. Therefore, the ROM of abduction and adduction is calculated by adding $\theta_{2}$ with $\theta_{4}$, and obtained 148° and 138.7° for males and females, respectively. The joint angles ($\theta_{5}$), which are most significantly involved in external and internal rotation movements, were 111.10° and 105.96° for males and females, respectively. Therefore, the ROM of external and internal rotation is calculated by $\theta_{5},$ and obtained 111.1° and 106° for males and females, respectively. In conclusion, the average ROM of ten subjects for abduction/adduction and external/internal rotation was, respectively, 143.4° and 108.6°. In abduction and adduction, males showed significantly higher ROM than females. Moreover, the elbow angle ($\theta_{6}$) of females was higher than males. Therefore, it is judged that females use the elbow more when moving in abduction and adduction than males to tracking motion trajectory. 

Unlike external and internal rotation, there are $\theta_{2}$ and $\theta_{4}$ that are centrally involved in the ROM of the shoulder in abduction and adduction. Besides $\theta_{2}$ and $\theta_{4}$, the rotation angles of $\theta_{3}$, $\theta_{5}$, and $\theta_{5}$ stand out. $\theta_{3}$ represents the left and right rotation of the shoulder. We think that this is likely due to the shoulder rotation working together with the help of the scapula during the rotation process of the shoulder. $\theta_{5}$ represents the rotation of the radius or ulna. In the posture of performing the initial movement, the direction vector of the palm faces the center of the body, but as ROM increases, it rotates outward and moves away from the center of the body. $\theta_{6}$ represents extension and flexion of the elbow. In the course of abduction and adduction exercise through motion capture, the exercise standard is 10 circular movements in a 180-degree range of motion. Therefore, in the process of exercising with the wrist in a half-moon-shaped orbit, if the ROM of the shoulder is limited, it is determined that the rotation follows the half-moon-shaped trajectory by flexion of the elbow.

Ropars classified shoulder hypersensitivity using a motion capture system and physical therapy goniometer. As a result, in the process of measuring standard data for the general public, the average ROM of the shoulder abduction and adduction was 129.9° ± 7.4°. Furthermore, the average external and internal rotational ROM of the shoulder was 94.3° ± 14.1 [24]. To analyze the scapular–humerus rhythm, Bagg, S. measured the movement of the scapula and humerus in abduction and adduction. As a result, the average ROM was 104.3° and a maximum movable range was 111.8° [25]. Barnes analyzed the ROM of shoulder movement using linear regression analysis and studied the age, gender, and dominance as comparative subjects. As a result, abduction and adduction was 180.1° ± 18.2 in males and 187.6° ± 16.1 in females, and the external and internal rotation was 101.2° ± 11.6 in males and 104.9° ± 12.0 in females. [26]. Rigoni validated an IMU for measuring shoulder range of motion in healthy adults. Each movement was assessed with a goniometer, and the IMU by two testers independently. Therefore, the compared agreements were assessed with intra-class correlation coefficients (ICC) and Bland–Altman 95% limits of agreement (LOA). As a result, the ROM of abduction and adduction were measured as 151.4 and 152.2, respectively; and internal and external rotation were measured as 141.1 and 142.3 with intra-class correlation (>0.90) [27]. 

The last thing to consider is the accuracy of the function of the motion capture (OptiTrack) system. Motion capture (OptiTrack) is very important for the accuracy of the sensor’s response when an object moves. Therefore, we can present the excellence of the accuracy of the proposed method by analyzing [28,29,30,31,32,33]. The method of this study and the methods of studies from [28,29,30,31,32,33] have different objects of observation and different quantities of sensors. However, since all of them used the same motion capture, it is possible to present an average value of accuracy for the function of motion capture. The average value for accuracy is recorded in Table 4, and it can be seen that the accuracy of this study improved by more than 10% compared to [28,29,30,31,32,33].

## 5. Conclusions

Based on the results of this study, the kinematics solution for 6-DOF of ROM could be determined through the standardized motion of the SR exercise, starting with the clavicle as the base point. In particular, based on the end effector information, we tried to solve the homogeneous transformation matrix of Equations (2) and (3) at first. However, because the constant values for the parameters (shoulder position) could not be solved, we changed the direction of the study to obtain the shoulder position first, then calculate the inverse kinematics formula. As a result, our approach differs compared to the commonly known 6-DOF inverse kinematics solution that combines 3-DOF of the wrist and the other joint angle of the 3-DOF. Through solving the 6-DOF inverse kinematics, future research and development of the 6-axis rehabilitation robot will be conducted. In the follow-up study, we will consider collecting the end effector data through force and torque sensors instead of using a motion capture system. If the end effector data is collected, the rehabilitation robot follows the trajectory of the patient’s motion through a kinematics solution. At the same time, the robot measures the maximum ROM of the patients, and it is envisaged that the patient will be able to perform stretching or passive or active-assisted exercises through the designated ROM.

Upper limb joints are structurally deeper than skin tissue, muscle tissue, and cartilage tissue. The existing research methods have limitations in objective evaluation because they do not select and analyze the central coordinates of the joint. However, because inverse kinematics can be automatically calculated by determining the center of the joint through motion capture, we think that it is extremely advantageous to suitably interpret the center coordinates of the joint, and the study results are more accurate and superior. Additionally, in order to reduce the standard deviation of the ROM and increase the accuracy of the experimental data, additional experiments should be conducted with increased subjects sample size.

If the cause of the difference in ROM is identified in the same rehabilitation motion, it is expected to make a great contribution to the analysis of rehabilitation exercise and human body mechanics. We think of two reasons why errors occur, even after repeating the same motion 10 times and obtaining an average ROM. The first is the degree of flexibility according to the ROM and muscle mass according to the patient’s own body shape. The second is judged to be a relative error of the center position according to the attachment position of the marker, even if it is purely the same operation. 

The movement of the SR robot must be the same as the human rehabilitation motion. Therefore, the proposed mathematical analysis method is sufficiently applicable because it is an analysis method for the objective evaluation of the movement of the rehabilitation robot. In conclusion, this study shows that a person will be able to exercise efficiently by wearing the rehabilitation robot with suggested kinematics model. Additionally, this study facilitated the determination of the ROM in the rehabilitation robot considering the ROM of normal subjects. Using the proposed model, it is possible to increase the accuracy of the trajectory of the rehabilitation robot and contribute to the improvement in safety. Comprehensively, utilizing the mathematical inverse kinematics equation that were debuted in this study, we will fabricate an upper rehabilitation robot through designing the mechanism instructor and motor in a follow-up study. In addition, because the rehabilitation exercise training-guided robot is linked to brain-related diseases, it contributes to the definition of domestic and international standardization of rehabilitation robots, affording universal training methods, accurate results, and objective evaluation for the safe treatment of patients.

## Figures and Tables

**Figure 1 diagnostics-12-03179-f001:**
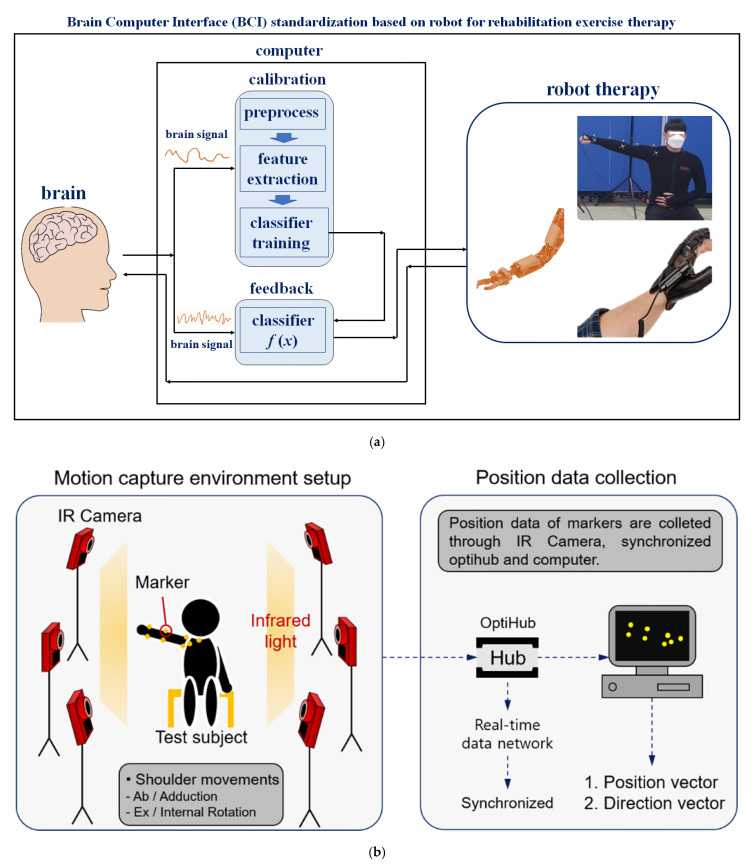
Configuration of a motion capture system for standardized rehabilitation exercise therapy. (**a**) Definition of the brain–computer interface (BCI). (**b**) Experimental environment setup for the motion capture and tracking markers.

**Figure 2 diagnostics-12-03179-f002:**
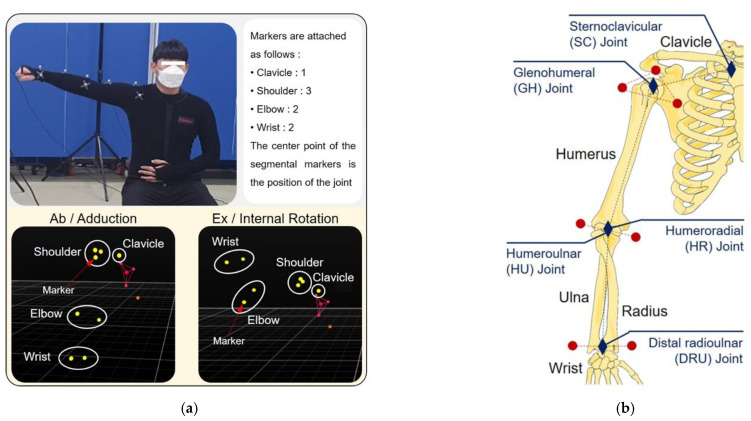
Photograph of the motion capture. (**a**) Abduction/adduction and external/internal rotation was performed to obtain the position and direction data of the markers. (**b**) The markers were attached to the skin to coincide with the central coordinate of the joint.

**Figure 3 diagnostics-12-03179-f003:**
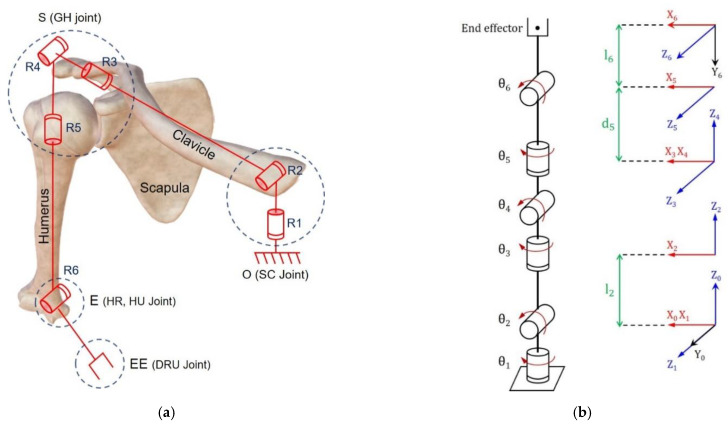
Shoulder complex modeling. (**a**) Mechanism of the shoulder complex model with rotation joints. (**b**) Forward kinematics modeling of the shoulder complex with a relative position coordinate system.

**Figure 4 diagnostics-12-03179-f004:**
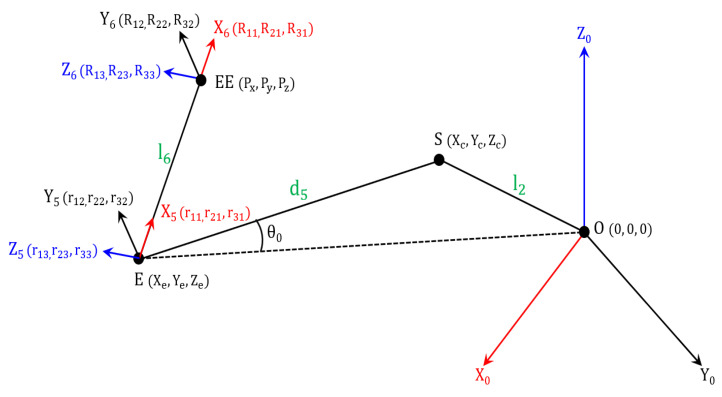
Position vector and direction vector of points E and EE.

**Figure 5 diagnostics-12-03179-f005:**
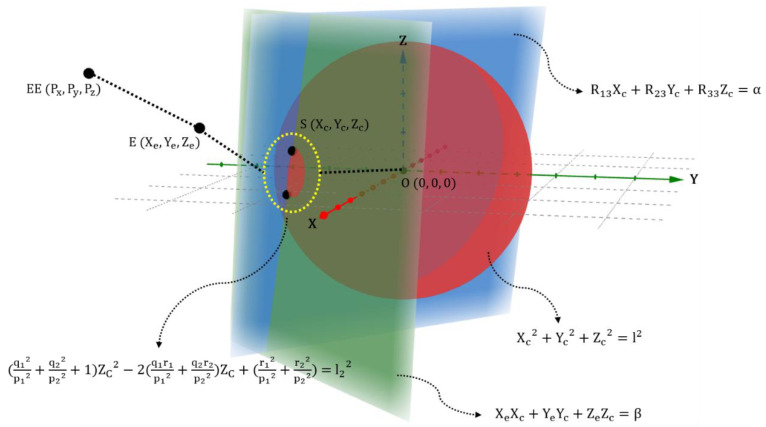
Two shoulder position vectors that are expressed through three-dimensional space.

**Figure 6 diagnostics-12-03179-f006:**
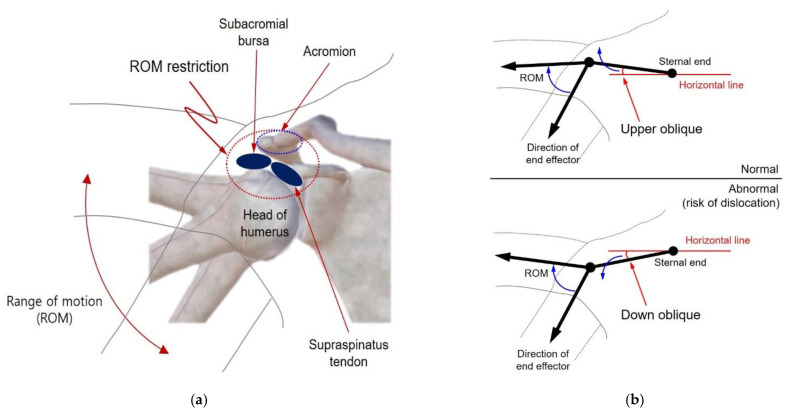
The biomechanical relationship between the shoulder and the acromioclavicular joint. (**a**) Anatomical structure of the shoulder joint and the acromioclavicular joint. (**b**) Normal or abnormal correlation of the inclination of the clavicle and shoulder rotation.

**Figure 7 diagnostics-12-03179-f007:**
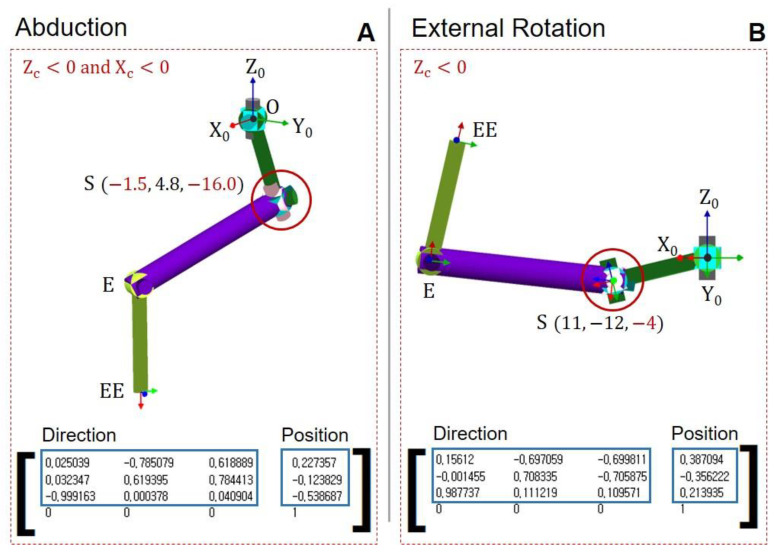
Simulation of shoulder movement based on the position vector of the end effector. (**A**) Math condition violation in abduction ($Z_{C}<0$ and $X_{C}<0$). (**B**) Math condition violation in external rotation ($Z_{C}<0$).

**Figure 8 diagnostics-12-03179-f008:**
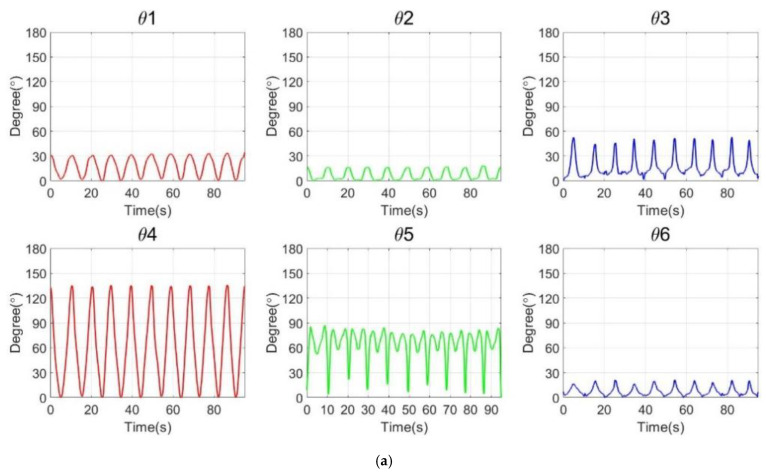
Joint rotation angle with ROM and simulation by analysis of the inverse kinematics. (**a**) Realized rotation degree variation. (**b**) Average ROM of males and females in abduction and adduction. (**c**) Simulation results of abduction and adduction. (**d**) Joint-centered trajectory graph in a 6-axis arm structure.

**Figure 9 diagnostics-12-03179-f009:**
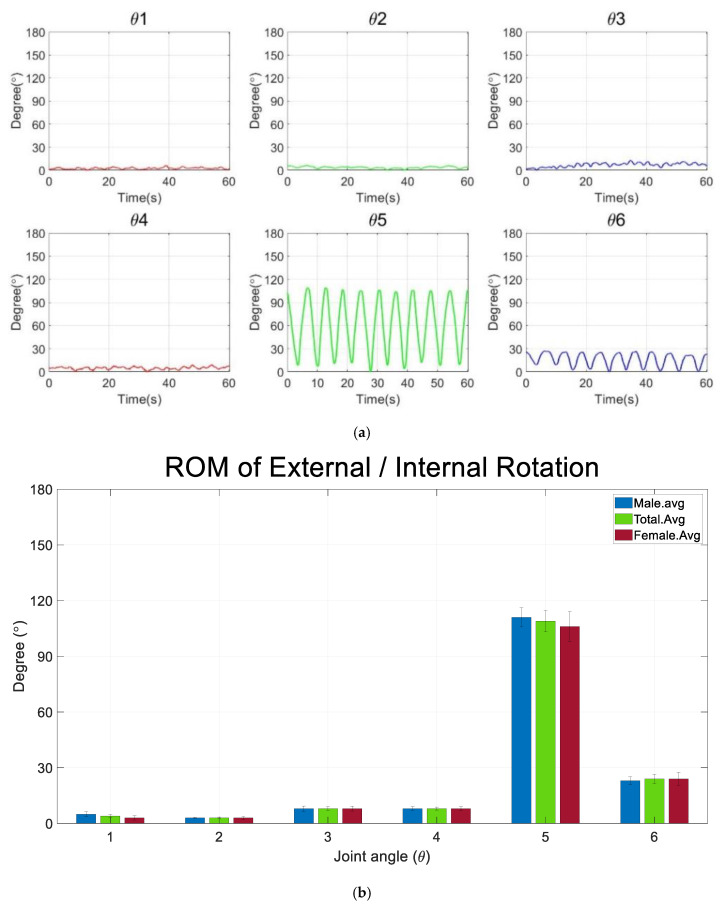
Joint rotation angle with ROM and simulation. (**a**) Realized rotation degree variation in external and internal rotation. (**b**) Average ROM for males and females. (**c**) Simulation results of external and internal rotation. (**c**,**d**) Simulation results of abduction and adduction. (**d**) Joint-centered trajectory graph in a 6-axis arm structure.

**Table 1 diagnostics-12-03179-t001:** Denavit–Hartenberg Table.

Joint	Link Angle $\theta_{i}\;(\mathrm{rad})$	Link Offset $d_{i}\;(\mathrm{mm})$	Link Length $l_{i}\;(\mathrm{mm})$	Link Twist $a_{i}\;(\mathrm{rad})$
1	$\theta_{1}$	0	0	$-\frac{\pi }{2}$
2	$\theta_{2}$	0	$l_{2}$	$\frac{\pi }{2}$
3	$\theta_{3}$	0	0	$-\frac{\pi }{2}$
4	$\theta_{4}$	0	0	$\frac{\pi }{2}$
5	$\theta_{5}$	$d_{5}$	0	$-\frac{\pi }{2}$
6	$\theta_{6}$	0	$l_{6}$	0

**Table 2 diagnostics-12-03179-t002:** Data collection for ROM (°) for males and females (abduction and adduction).

	Height (mm)	$\theta_{1}$	$\theta_{2}$	$\theta_{3}$	$\theta_{4}$	$\theta_{5}$	$\theta_{6}$
M._avg_	177	28.9	17.6	46.1	130.4	69.3	20.7
M_1_	170	33.1	18.1	44.5	127.3	79.9	26.4
M_2_	174	30.8	15.9	45.3	133.7	72.0	18.3
M_3_	177	24.8	25.9	43.9	130.4	69.8	24.0
M_4_	190	29.1	14.9	33.8	134.3	54.8	10.5
M_5_	172	26.8	13.4	63.1	126.4	70.1	24.5
SD	7.1	2.9	4.4	9.5	3.2	8.1	5.8
SEM	3.2	1.3	2.0	4.2	1.4	3.6	2.6
F_.avg_	160	18.3	11.5	31.9	127.2	44.8	26.2
F_1_	161	21.5	11.3	26.0	132.9	48.2	24.9
F_2_	159	14.4	12.4	25.8	116.1	23.5	33.7
F_3_	165	13.5	6.6	33.1	108.2	63.6	28.0
F_4_	160	21.7	14.8	45.2	146.4	42.5	14.5
F_5_	154	20.3	12.5	29.2	132.5	46.3	29.9
SD	3.5	3.6	2.7	7.2	13.5	12.9	6.5
SEM	1.6	1.6	1.2	3.2	6.0	5.7	2.9
T._avg_		23.6	14.6	39.0	128.8	57.1	23.5
SEM		2.0	1.5	3.5	3.1	5.2	2.1

SD (σ): Standard deviation. SEM: Standard error of the mean. T._avg_: Total average.

**Table 3 diagnostics-12-03179-t003:** Data collection for the ROM (°) for males and females (external and internal rotation).

	Height	$\theta_{1}$	$\theta_{2}$	$\theta_{3}$	$\theta_{4}$	$\theta_{5}$	$\theta_{6}$
M._avg_	177	4.9	3.2	8.0	8.5	111.1	23.4
M_1_	170	2.6	3.4	8.5	9.4	109.9	18.8
M_2_	174	2.8	2.9	3.5	3.9	97.5	22.1
M_3_	177	2.0	2.2	5.4	9.3	99.5	20.6
M_4_	190	9.0	4.4	11.5	10.2	125.9	32.1
M_5_	172	8.0	3.3	11.2	9.5	122.7	23.6
SD	7.1	3.0	0.7	3.2	2.3	11.6	4.6
SEM	3.2	1.3	0.3	1.4	1.0	5.2	2.1
F_.avg_	160	2.9	3.4	7.9	7.5	106.0	23.6
F_1_	161	6.6	6.9	12.3	8.8	138.5	42.9
F_2_	159	1.3	1.0	5.6	5.1	79.7	22.3
F_3_	165	1.2	1.8	5.7	4.4	89.9	19.2
F_4_	160	3.4	4.5	7.5	8.4	122.4	19.5
F_5_	154	2.0	3.0	8.4	10.9	90.3	13.9
SD	7.1	2.0	2.1	2.4	2.4	22.4	10.0
SEM	3.2	0.9	0.9	1.1	1.1	10.0	4.5
T._avg_		3.9	3.3	8.0	8.0	107.6	23.5
SEM		0.9	0.5	0.9	0.8	5.7	2.5

SD (σ): Standard deviation. SEM: Standard error of the mean. T._avg_: Total average.

**Table 4 diagnostics-12-03179-t004:** Comparison for accuracy of proposed system and others.

Reference	Average Accuracy [%]	Performance of a Motion Capture
this work	97.6	OptiTrack
[28]	94.8	optical motion capture method (DeepMoCap)
[29]	93.6	multi-person pose estimation
[30]	95.9	OptiTrack
[31]	95.0	IMU Sensor (mobilitylLab system)
[32]	85.3	multiple Kinect sensors
[33]	70.0	kinect V2 and captiv sensor

## Data Availability

The data presented in this study are available upon request from the corresponding author. The data are not publicly available because of privacy and ethical re-strictions.

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
