# Peer review of "Mathematical Analysis and Motion Capture System Utilization Method for Standardization Evaluation of Tracking Objectivity of 6-DOF Arm Structure for Rehabilitation Training Exercise Therapy Robot"

_diagnostics, 2022, doi:10.3390/diagnostics12123179_

Round 1
Reviewer 1 Report
1. Is the kinematic forward and inverse solution between the two points
shown in the article a starting point for rehabilitation training? If
so, how will patient safety be ensured during the transition? Also, is
the actual check done instead of staying in the simulation results?
2. The joint center points of different patients are inconsistent. How
to apply the same set of equipment to different groups of people? In
addition to angular considerations, are there lengths between links that
need to be considered?
3. What is the significance of brain-computer interface? The article
needs to be detailed.
4. Can the sample size be increased to improve the persuasiveness of the
data?
Author Response
1. Q. Is the kinematic forward and inverse solution between the two points shown in the article a starting point for rehabilitation training? If so, how will patient safety be ensured during the transition? Also, is the actual check done instead of staying in the simulation results?
A. Thank you for your valuable advice on our manuscript. As forward and inverse kinematics were calculated to obtained the rotation angle of each joint, the study direction was chosen to measure the ROM (range of motion) of normal subject and utilize the inverse kinematics mathematical solution for rehabilitation robot. [Line : 109 - 112]
In particular, prior to obtain joint rotation angle, we express the research method by designating the two point that indicateds position coordinates of the elbow and shoulder during processing of the inverse kinematics solution. [Line : 320-322]
Utilizing mathematical inverse kinematics equation that were soluted in this study, we will make upper rehalibitation robot through designing the mechanisim instructor and motor as a follow-up study. Thanks to suggedsted comments, we were able to add this comments to the discussion section. [Line : 533-535]
2. Q. The joint center points of different patients are inconsistent. How to apply the same set of equipment to different groups of people? In addition to angular considerations, are there lengths between links that need to be considered?
A. Thank you for your valuable advice on our manuscript. All subejcts performed the experiment by wearing the same flexible suit and attaching the marker. In particular, we attach the marker location to designate the center of joint through calculating the average position formula of the triangle and line. During attaching the marker, physical therapist advised to author for chossing the precise joint center location. [Line : 162-165]
Links that representing the lengths of two different center joint position (eg. Humerus or radius) were different from subjects, it can be calculated through the distance formula between two points on 3-dimensional space. In this study, we used matlab tool to derived the inverse kinematics solution by reflecting the links that change in real time. Additionally, mathmetical equation was added and modified in the method section based on this comments. [Line : 203-209]
3. Q. What is the significance of brain-computer interface? The article needs to be detailed.
A. Thank you for your valuable advice on our manuscript. Quadriplegic patients, deaf, blind, and speech impaired cannot express themselves accurately to therapists during exercise training programs for rehabilitation treatment [17]. Therefore, if emergencies and emergencies occur in the course of training and treatment using treatment instruments, the therapist does not recognize the patient's condition and a medical accident occurs. Brain Computer Interface (BCI) defines a technology for inter-action between the brain and computer [18]. This technology refers to a control technology that provides a service so that a computer can grasp the thoughts intended by humans and move objects [19]. In other words, BCI detects brain waves so that computers can grasp cognition, learning, and reasoning like the human brain [20]. Therefore, it is ana-lyzed that the use of BCI technology will be high for quadriplegic patients, hearing im-paired, visually impaired, and speech impaired who need rehabilitation exercise. BCI technology uses a camera to capture the movement of the patient through the camera, and assistively reads the EEG and analyzes the data obtained from the camera and EEG diag-nosis to identify the patient's movement pattern. Therefore, it is possible to predict the treatment outcome by understanding the patient's requirements and condition. We wrote this contents in the Analysis of Motion Capture secrion. [Line : 118-132]
4. Q. Can the sample size be increased to improve the persuasiveness of the data?
A. Thank you for your valuable advice on our manuscript. It's really unfortunate, but it is currently impossible to offer an experiment place because of research deadline. We also think that as the sample size increases, the variance of ROM and standard deviation will be decreased and the accuracy of experiment data will increase. Therefore, the content was added to the discussion section, and additional experiments will be conducted in the process of designing a rehabilitation robot in the follow-up study. [Line : 543-546]

Reviewer 2 Report
please see the attached file.
This paper presents a method capable of computing joint angles of the human arm using a motion capture system and inverse kinematics of a rigid body arm for rehabilitation training purposes. The following comments should be taken into account and answered properly to eliminate scientific and technical doubts about the method and clarify the contributions of the paper.
1. In the reviewer’s opinion, the contribution of this paper is not very clear. The authors used several markers rather than one marker to estimate joint centers of the shoulder, elbow, and wrist (Figure 2), and in this case, the reviewer thinks that it just increases the difficulty in finding the correct position of each marker on bones due to soft skin and muscles as well. Authors need to show how much the usage of 2 or 3 markers improves in finding the joint center over the usage of 1 marker (quantitatively). Additionally, the use of the rigid body arm model doesn’t represent the human arm because the rotational axis of the arm joints is not fixed but shifted, which is known to impede an exact alignment of a wearable rehabilitation robot with the arm.
2. The paper proposes a method for analyzing shoulder movement and shows the results with abduction/adduction and external/internal rotation. Then why do the authors use a 6-DOF arm structure? For example, the elbow joint is really necessary? Moreover, even precisely finding the rotational axes of 3 DOF is a difficult task because they are shifted according to shoulder motion. In the reviewer’s opinion, identifying such shifting would have a large contribution.
3. The abastact shows that Abduction was 180.1°±18.2 in males and 187.6°±16.1 in females, and external rotation was 101.2°±11.6 in males and 104.9°±12.0 in females. The variation is almost 10% of the mean which is so high. What causes such variation? Due to large variation in the anthropometric data of the subjects? Please explain where such variations come from and which joint.
4. Please add more existing studies about human motion analysis related to the paper. There are several methods, for example, based on IMUs, etc. Additionally, the reviewer suggests t show quantitative (also qualitative) comparison results with other methods to show the pros and cons of the proposed method over the existing methods. It will be helpful to justify that the proposed method is more suitable than others as a standardized method.
5. In line 142, why the suit is used? This kind of suit is barely applicable to patients for rehabilitation.
6. As regards Comment 2, please show the advantages of the model in Fig 3 (a) over the smaller DOFs model.
7. In lines 195-196, it is true that the lengths of the upper and lower arms vary with the patient but each patient has constant lengths. Usually, they are measured before performing the experiments. What is the benefit of real-time computing over fixed values? How to calculate them? If variation appears, how much is it as compared to the fixed value?
8. The contents of Pages 6, 7, and 8 (lines 188-232) are shown in a common Robotics textbook so such details are not necessary for the paper. Table 1, equations (2)-(3), and a short description would be enough.
9. In Figure 4, please specify S (Xc, Yc, Zc). The reviewer doesn’t understand the main purpose of Subsection 3.2.1. The rotational axis of the shoulder could be directly calculated by the markers’ data that are placed around the shoulder. Moreover, if the position of the rotational axis of the shoulder is not determined, then how can the authors calculate the equations (2)-(3) that require information on the position of the shoulder joint?
10. Please correct the numbering of Joint angle analysis section (Page 12). It should be 3.2.2.
11. Does Figure 8 show averages of 10 trials or just one trial? If it is average, please also add the deviation to the graph. The reviewer proposes to show numerical values of the position of the end-effector (also external and internal rotation).
12. As the graph shows abduction and adduction of the shoulder, in principle, the shoulder angles should be changed dominantly but it looks like the other angles also show large changes. Please explain why.
13. As all angles are zeros in the first figure of Figures 8 (c) and 9 (c), their configuration should be the same. Why are they different?
14. Regarding the results of Section 4, please also show the movement of the arm using makers’ data and compare it with the movement obtained by the simulator.
15. Although Tables 4 and 5 show the changes of the ROM for different szie subjects, showing why the difference of ROM occurs while performing the same movement would have a large contribution. In the reviewer’s opinion, the average data is not useful for rehabilitation because the patient needs the training to cover specific ROM adapted to the patient.
16. In lines 492-495, please make a discussion of why such reverse results were obtained. Additionally, the reviewer proposes to add a discussion section to explain the above comment including the limitation of the study, etc.
17. In line 507, ‘(motion)’ seems useless. Error or something missing?
18. Additional discussion and analysis will be needed to justify the texts on Page 20.

Author Response
1. Q. In the reviewer’s opinion, the contribution of this paper is not very clear. The authors used several markers rather than one marker to estimate joint centers of the shoulder, elbow, and wrist (Figure 2), and in this case, the reviewer thinks that it just increases the difficulty in finding the correct position of each marker on bones due to soft skin and muscles as well.
Authors need to show how much the usage of 2 or 3 markers improves in finding the joint center over the usage of 1 marker (quantitatively). Additionally, the use of the rigid body arm model doesn’t represent the human arm because the rotational axis of the arm joints is not fixed but shifted, which is known to impede an exact alignment of a wearable rehabilitation robot with the arm.
A. Thank you for your good comments. We also agree that the method suggested by the reviewer is a very good method and increases accuracy. We attached 8 markers in our first experiment. The reason was to designate the center point of several markers as the center axis of joint rotation, On the other hand, using one marker has the advantage of simplifying the patient's kinematic analysis and simplifying the test. However, since joints are surrounded by skin, muscle and cartilage, locating one marker for each joint can be challenging. So even though we used 8 markers (1 clavicle, 3 shoulders, 2 elbows, 2 wrists), I think we need to do a comparative test using 1 marker.
Since the experiment for the current study has been completed, it is necessary to recruit subjects and replan the research environment for follow-up research. In the follow-up study of rehabilitation robots, if a test is performed using one marker and comparative analysis of accuracy is performed, it is likely that greater results will be obtained in analyzing the mechanical movement of the human upper limb. Thank you so much for your sincere advice.
2. Q. The paper proposes a method for analyzing shoulder movement and shows the results with abduction/adduction and external/internal rotation. Then why do the authors use a 6-DOF arm structure? For example, the elbow joint is really necessary? Moreover, even precisely finding the rotational axes of 3 DOF is a difficult task because they are shifted according to shoulder motion. In the reviewer’s opinion, identifying such shifting would have a large contribution.
A. Thank you for your valuable review. The purpose of this study is to measure ROM in the case of abduction and adduction, external and internal rotation. Moreover, the 6-DOF structure was solved to additionally analyzed inverse kinematics that was used in the follow-up research and development (R&D) of upper limb rehabilitation robots. As the reviewer said, the rotation angle of the elbow joint has little effect on the shoulder ROM in abduction and adduction, external and internal rotation. However, 6-axis inverse kinematics was solved to prepare for follow-up R&D of the robot, receiving the end effector data that was collected from the force / torque sensor on the wrist and following the movement of patient in real time. In this study, the base point, the positional coordinates of each joint and the inverse kinematics solution for the end effector were calculated using Matlab tool based on the motion capture system. Regarding the content, we modified how to utilize 6-DOF in discussion section. Thank you so much for your sincere advice. [Line : 535-537]
3. Q. The abstract shows that Abduction was 180.1°±18.2 in males and 187.6°±16.1 in females, and
sadfexternal rotation was 101.2°±11.6 in males and 104.9°±12.0 in females. The variation is almost 10% of the mean which is so high. What causes such variation? Due to large variation in the anthropometric data of the subjects? Please explain where such variations come from and which joint.
A. Thank you for your valuable review. We mistakenly attached the contents in the process of writing the abstract. The information about ROM in abstract that you read is the result of Barnes study [22]. The author derived the following research results from this paper and revised the abstrack content. ROM of the abduction and adduction were 148° in males and 138.7° in females. Moreover, ROM of the external and internal rotation were 111.2° in males and 106° in females. The ROM is the only ROM result of the shoulder exercise, excluding the ROM of the elbow. Thank you so much for your sincere advice. [Line : 34-39]
4. Q. Please add more existing studies about human motion analysis related to the paper. There are several methods, for example, based on IMUs, etc. Additionally, the reviewer suggests show quantitative (also qualitative) comparison results with other methods to show the pros and cons of the proposed method over the existing methods. It will be helpful to justify that the proposed method is more suitable than others as a standardized method.
A. Thank you for your valuable advice. Previous research on upper extremity kinematics measurement methods was written in the introduction, and the results were compared and analyzed on IMU and geometry, and ROM measurement studies through motion capture were modified and supplemented. In particular, when comparing ROM using IMU and geometry for 30 normal subjects, it was confirmed that they had a high ICC score (>0.90) [23]. The contents were summarized and added to conclusion section Thank you so much for your sincere advice. [Line : 516-522]
5. Q. In line 142, why the suit is used? This kind of suit is barely applicable to patients for rehabilitation.
A. Thank you for your interest and questions. The suit is worn to freely allow the marker to be attached and detached. To coincides position of joint center with the straight line and triangle that formed by the marker, markers are attached on the skin. Due to the word selection in the line, ambiguous sentences were modified in the manuscript. [Line : 152 and 159-161]
6. Q. As regards Comment 2, please show the advantages of the model in Fig 3 (a) over the smaller DOFs model.
A. Thank you for your interest and questions. Fig. 3(a) shows the joint configuration of the upper limb defined based on human anatomical motion by showing a 6-DOF rotational joint. It consists of 2 axes of left/right and up/down rotation of the clavicle, 3 axes of roll, pitch, and yaw rotation of the shoulder, and 1 axis of flexion and extension of the elbow. Even with a degree of freedom less than 6-DOF, the end effector can be moved to the desired position according to the modeling of the rotary joint. However, as the DOF gradually increases, it can show a more fine and smooth trajectory than a model with fewer DOF. Therefore, the mathematical solution of the 6-DOF model shows better resolution in tracking the motion of the human body than the relatively small DOF model. Additionally, as the DOF increases, various value of the joint anglesare calculated according to the position tracking of the end effector. Therefore, relatively complex formulas and conditions are required versus less DOF complex, but nonetheless, it is used because it can represent the most similar motion to the human body.
7. Q. In lines 195-196, it is true that the lengths of the upper and lower arms vary with the patient but each patient has constant lengths. Usually, they are measured before performing the experiments. What is the benefit of real-time computing over fixed values? How to calculate them? If variation appears, how much is it as compared to the fixed value?
A. Thank you for your interest and questions. For the solution of inverse kinematics, the appropriate matrix value is derived by applying the DH parameter theory when solving forward kinematics in advance. In particular, each person have the different link length (l_n and d_n) in the homogeneous transformation matrix. Therefore, after deriving the position coordinates of the shoulder, elbow, and wrist, the formula for obtaining the link length is substituted into Matlab tool. As a result, the link length that changes in real time is applied to solve the inverse kinematics. The authors substitute the link (l_n and d_n) that solved formula applicable to this content into the manuscript, and the average link length (l_2, d_5 and l_6) are 167.6mm, 306.5mm and 214.5mm at clavicle, humerus and radius. In order to make the accurate context and reflect the reviewer's comments, we delete the contents of lines (195-196) and the contents of Equation (1) were added to the Forward kinematics section. [Line : 203 – 209]
8. Q. The contents of Pages 6, 7, and 8 (lines 188-232) are shown in a common Robotics textbook so such details are not necessary for the paper. Table 1, equations (2)-(3), and a short description would be enough.
A. Thank you for your valuable advice. While working hard on the paper, it seems that we started writing the basic contents to give readers specific information. The content of the forward kinematics section was written with the intention of explaining which points are taken as base point and end effectors when analyzing kinematics for shoulder movement using motion capture. Following the reviewer's advice, the text was summarized as much as possible and modified to add only necessary explanations. [Line : 178-224]
9. Q. In Figure 4, please specify S (Xc, Yc, Zc). The reviewer doesn’t understand the main purpose of Subsection 3.2.1. The rotational axis of the shoulder could be directly calculated by the markers’ data that are placed around the shoulder. Moreover, if the position of the rotational axis of the shoulder is not determined, then how can the authors calculate the equations (2)-(3) that require information on the position of the shoulder joint?
A. Thank you very much for asking questions about deep and in-depth content. In this study, we present a 6-DOF inverse kinematics solution starting with the clavicle as the base point. In particular, in the process of solving 6-axis inverse kinematics based on end effector information, the homogeneous transformation matrix of equations (2) and (3) must be calculated first as the reviewer said. However, in the process of solving, constant values for parameters (shoulder position) could not be solved. Therefore, we decided the direction of the study that after obtaining the position information of the shoulder first, calculate the joint angle through the inverse kinematics formula. We added to the discussion section that we took a different approach to this content, compared to the commonly known 6-DOF inverse kinematics solution that combines 3-DOF of the wrist and the other joint angle of the 3-DOF. Thank you very much.
[Line : 527-533]
10. Q. Please correct the numbering of Joint angle analysis section (Page 12). It should be 3.2.2.
A. Thank you for your valuable advice. The number of the “Joint Angle Analysis” section has been modified to 3.2.2, and grammar and syntax errors have been corrected through an additional overall review. [Line : 323]
11. Q. Does Figure 8 show averages of 10 trials or just one trial? If it is average, please also add the deviation to the graph. The reviewer proposes to show numerical values of the position of the end-effector (also external and internal rotation).
A. Thank you for your valuable advice. Figure 8 (a) is the joint angle pattern during 10 repetitive movements of one subject. Table 2 (abduction and adduction) and Table 3 (external and internal rotation) show the average ROM during the 10 repetitions of each experimenter. The table 2 and 3 show the average ROM for each rotational joint when repeated exercise was performed. For the average ROM by rotation angle of 10 subjects, the standard deviation (σ) was added to Table 2 and Table 3, respectively. Additionally, in the simulations of Figures 8 and 9 (c), we added the end effector values in the moving section based on the initial position. [Line : 417-419, figure 8(c) and figure 9(c)]
12. Q. As the graph shows abduction and adduction of the shoulder, in principle, the shoulder angles should be changed dominantly but it looks like the other angles also show large changes. Please explain why.
A. Thank you for your valuable advice. Unlike external and internal rotation, there are θ_2 and θ_4 that are centrally involved in the ROM of the shoulder in abduction and adduction. Besides θ_2 and θ_4, rotation angle of θ_3, θ_5 and θ_5 were stand out. θ_3 represents the left and right rotation of the shoulder. We think that it is analyzed that because the shoulder rotation works together with the help of the scapula rithm during the rotation process of the shoulder. θ_5 represents the rotation of the radius or ulna. In the posture of performing the initial movement, the direction vector of the palm faces the center of the body, but as ROM increases, it rotates outward and moves away from the center of the body. θ_6 represents extension and flexion of the elbow. In the course of abduction and adduction exercise through motion capture, the exercise standard is 10 round trips in a 180-degree range of motion. Therefore, in the process of exercising with the wrist in a half-moon-shaped orbit, if the ROM of the shoulder is limited, it is determined that the rotation occurred to follow the half-moon-shaped trajectory by flexion the elbow. Thanks to the reviewers, we were able to add an analysis of the results to the conclusion section. [Line : 493-505]
13. Q. As all angles are zeros in the first figure of Figures 8 (c) and 9 (c), their configuration should be the same. Why are they different?
A. Thank you for your interest in our manuscript. In Figures 8 (c) and 9 (c), the authors set the initial position as 0 based on the amount of rotation angle change to represent the ROM simulation. However, we confirmed that the author's interpretation was wrong and caused misunderstanding to reviewer. Therefore, the rotation angle of each joint is added in the each figures, based on the initial position of 0° of inverse kinematics. At the same time, the homogeneous transformation matrix component of the end effector is also attached. Line : figure 8(c) and figure 9(c)]
14. Q. Regarding the results of Section 4, please also show the movement of the arm using maker's’ data and compare it with the movement obtained by the simulator.
A. Thank you for your valuable comments. The author tracked the position of each joint in the right arm using motion capture data in abduction and adduction. Therefore, we found one difference between the robot simulation and movement of the human. Simulation gives a certain angle to form repetitive ROM in one scaption line, but in the movement based on motion capture data, ROM was obtained through repetitive motion in various scaption lines. As a result, it was found that it is important not only to measure ROM through the rotation joint, but also to track and analyze the subject's movement trajectory. We wrote this one difference on the result section to notice all reviewer. Thank you for your advise. [Line : 429 – 437 and 473-477]
15. Q. Although Tables 4 and 5 show the changes of the ROM for different szie subjects, showing why the difference of ROM occurs while performing the same movement would have a large contribution. In the reviewer’s opinion, the average data is not useful for rehabilitation because the patient needs the training to cover specific ROM adapted to the patient.
A. Thanks for the advice on our manuscript. As the reviewer said, if the cause of the difference in ROM is identified, it is expected to make a great contribution to the analysis of rehabilitation exercise and human body mechanics. We think of two reasons why errors occur even after repeating the same motion 10 times and obtaining an average ROM. The first is the degree of flexibility according to the ROM and muscle mass according to the patient's own body shape. The second was judged to be a relative error of the center position according to the attachment position of the marker even if it is purely the same operation. We want to share it with reviewers or readers by adding it to the discussion section. [Line : 547-553]
16. Q. In lines 492-495, please make a discussion of why such reverse results were obtained. Additionally, the reviewer proposes to add a discussion section to explain the above comment including the limitation of the study, etc.
A. Thanks for the advice on our manuscript. In this study, shoulder abduction and adduction of males have higher ROM than females. Also, external and internal rotation of males also have higher ROM than females. In this case, abduction and adduction use the rotation joint that consisted by θ_2 and θ_4 angle, predominantly. Additionally, external and internal rotation use the rotation joint that consisted by θ_5. In conclusion, θ_2, θ_4 and θ_5 is more used with males that females. However, females have more joint rotation angle (θ_6) than males. Therefore, we judged that female more use the elbow moving in abduction and adduction than male to tracking motion trajectory. We wrote the contents with quantification ROM in the conclusion section. [Line : 480-492]
17. Q. In line 507, ‘(motion)’ seems useless. Error or something missing?
A. Thanks for the advice on our manuscript. It is an error that occurred in the process of writing this menuscript. Thank you for adivce to our. The word ‘motion’ was deleted. Furthermore, the contents of the conclusion were completely revised to constitue the precise and depth sentence. [Line : 525-527]
18. Q. Additional discussion and analysis will be needed to justify the texts on Page 20.
A. Thank you for your valuable advice. In order to discuss ROM measurement using motion capture, supplements and improvements, and to discuss future directions for the application of upper limb rehabilitation robots using the proposed formula, the conclusion and discussion sections were separated and modified. This study does not end here, but as a follow-up study through the process of manufacturing rehabilitation robots, it is believed that the analytical accuracy for inverkse kinematics using motion capture will be steadily supplemented.

Round 2
Reviewer 2 Report
please find the attached file

Author Response
Q 1. Please switch the order of the discussion and conclusion sections. The conclusion section should be last
A 1. Thank you for your good comments. The meaning of the aritcles written in the discusion and conclusion section is the same. Therefore, we only swithced the section title of the discussion and conclusion. [Line : 494, 548]
Q 2. Tables 2 and 3 contain the standard deviation. Please separate it for male and female subjects and also add this info in Figure 8 (b) and Figure 9 (b).
A 2. Thank you for your valuable review. The authors presented the standard deviation and standard error of the mean separately for males and females to represent accurate and objective evaluation indicators. Additionally, we added the ROM of the total average and the corresponding indicator was added to the figure 8 (b) and figure 9 (b). [Line : Table 2, 3 and Figure 8 (b) and 9 (b)]
Q 3. As the authors mentioned, the reviewer strongly recommends to show that the comparison results of the proposed method over the conventional methods in the following studies and papers.
A 3. Thank you for your valuable review. Motion capture (optitrack) is very important for the accuracy of the sensor's response when an object moves. Therefore, we can present the the accuracy of the proposed method by analyzing previous studies. The method of this study and the methods of previous studies have different objects of observation and different quantities of sensors. However, since all of them used the same motion capture, it is possible to present an average value of accuracy for the function of motion capture. The average value for accuracy is recorded in Table 4 that was written in the discussion section. Once again thank you for your sincere advice. [Line : 538-547]
Q 4. The proposed method will be integrated into SR robot? For example, real-time calculation of inverse kinematics to define joint angles of SR robot during the training? Or first, get the joint angle data that the patient could perform, then SR robot follows the predefined angles? This topic needs to be addressed in the discussion section.
A 4. Thank you for your valuable advice. When patient wears the robot, we will measure the maximum ROM of the patients, using real-time inverse kinematics calculation in the first step. In the next step, the patient will perform passive, stretching or active-assisted exercises through the designated ROM. We added the this contents at the conclusion section. [Line : 562-564].
Q 5. In any case, the use of such a suit is not recommended for the patient. In addition, attaching the markers on the body with SR robot (if it is a wearable type) might not be easy. This topic needs to be addressed in the discussion section.
A 5. Thank you for your sincere advice. In fact, we used a motion capture system to analyze the movements of abduction / adduction and external rotation / internal rotation, which are standard for rehabilitation exercises, and to analyze the results of inverse kinematics in the research. In the follow-up study on rehabilitation robots, we are considering excluding the motion capture system and the suit in order to receive the end-effector through the 6-axis torque sensor and control the movement of the robots. We added the this contents in the conclusion section. [Line : 559-562]
